# Development of an AI-Based Predictive Algorithm for Early Diagnosis of High-Risk Dementia Groups among the Elderly: Utilizing Health Lifelog Data

**DOI:** 10.3390/healthcare12181872

**Published:** 2024-09-18

**Authors:** Ji-Yong Lee, So Yoon Lee

**Affiliations:** 1Center for Sports and Performance Analysis, Korea National Sport University, Seoul 05541, Republic of Korea; 302479@knsu.ac.kr; 2Department of Physical Education, Korea National Sport University, Seoul 05541, Republic of Korea

**Keywords:** artificial intelligence, early dementia detection, lifelog data, wearable devices, machine learning

## Abstract

Background/Objectives: This study aimed to develop a predictive algorithm for the early diagnosis of dementia in the high-risk group of older adults using artificial intelligence technologies. The objective is to create an accessible diagnostic method that does not rely on traditional medical equipment, thereby improving the early detection and management of dementia. Methods: Lifelog data from wearable devices targeting this high-risk group were collected from the AI Hub platform. Various indicators from these data were analyzed to develop a dementia diagnostic model. Machine learning techniques such as Logistic Regression, Random Forest, LightGBM, and Support Vector Machine were employed. Data augmentation techniques were applied to address data imbalance, thereby enhancing the model performance. Results: Data augmentation significantly improved the model’s accuracy in classifying dementia cases. Specifically, in gait data, the SVM model performed with an accuracy of 0.879. In sleep data, a Logistic Regression was performed, yielding an accuracy of 0.818. This indicates that the lifelog data can effectively contribute to the early diagnosis of dementia, providing a practical solution that can be easily integrated into healthcare systems. Conclusions: This study demonstrates that lifelog data, which are easily collected in daily life, can significantly enhance the accessibility and efficiency of dementia diagnosis, aiding in the effective use of medical resources and potentially delaying disease progression.

## 1. Introduction

The escalating threat of dementia in aging societies poses a significant social and economic burden. As the global population ages, the number of people with dementia continues to increase. Dementia is a common geriatric condition affecting approximately 10% of individuals aged 65 years and above [1]. It denotes a state characterized not only by a decline in cognitive function but also by impairments in language, intelligence, concentration, and judgment abilities, indicating anomalies in perceptual skills. Once dementia develops, it is not reversible and tends to either remain stable or progressively worsen [2]. The disease places a substantial burden not only on the patients themselves but also on their family members, necessitating proactive national responses [3]. Despite the various types and symptoms of dementia, its underlying mechanisms remain unclear, and no definitive treatments are currently available. Consequently, early diagnosis of dementia is crucial and increasingly emphasized [4,5].

Globally, the number of individuals diagnosed with dementia continues to rise steadily. According to the World Health Organization (WHO), it is projected that by 2050, the number of dementia patients will reach approximately 152.8 million, which is nearly three times the current figure [6]. The economic burden of dementia is significant; in 2019, the global cost for 55.2 million individuals with dementia was estimated at USD 1.313, equating to an average cost of USD 23,796 [7]. This financial burden is expected to increase alongside the aging population, highlighting the critical need for the establishment of early diagnosis systems for dementia.

Early diagnosis is the most effective method for managing dementia, allowing for interventions to delay its progression to severe stages. Traditionally, the diagnosis of dementia relies on the clinical expertise of physicians and often involves expensive neuroimaging assessments [8]. This makes the early diagnosis of dementia difficult and inaccessible for many individuals. National-level initiatives, such as the Mini-Mental State Examination for Dementia Screening conducted at public health centers, aim to address this issue [9]. However, delays in acknowledging symptoms often prevent timely visits to health centers for diagnosis.

If dementia is not diagnosed early, patients may miss the critical treatment window, losing a valuable opportunity to slow the progression of the disease. As previously mentioned, dementia is a progressive disorder that worsens over time. Without appropriate treatment and management during the early stages, the disease can quickly advance to severe stages [2]. Failure to diagnose early means that patients may not receive the necessary treatment until the disease has significantly progressed, leading to accelerated functional decline and an increased risk of losing independence in daily activities. Additionally, in the early stages of dementia, many patients either do not notice the decline in cognitive function or dismiss mild symptoms, resulting in a diagnosis only after the disease has advanced considerably. This situation often increases the psychological and financial burden on both the patients and their families, significantly diminishing the quality of life for those affected by dementia. Therefore, early detection and proactive treatment are essential for maintaining patient functionality and improving quality of life.

Traditional dementia diagnosis methods face two main challenges: the need for individuals to visit diagnostic facilities such as hospitals or health centers, and the reliance on expensive equipment [10]. Which solutions address these issues? A potential solution for early diagnosis is to record data generated from daily life activities utilizing equipment that is readily available, thereby overcoming reliance on expensive diagnostic tools [11].

Lifelog data generated through Internet of Things (IoT) technologies, such as wearable devices and mobile equipment, provides a comprehensive record of an individual’s daily life activities. Although lifelog data includes information recorded on social networking sites, their most noteworthy application is in the healthcare industry [12]. In healthcare, notable lifelog data include activity levels, sleep information, dietary habits, weight fluctuations, body mass index, and muscle mass data collected from smartphones and wearable devices [13]. The healthcare industry aims to leverage these data to address weaknesses in medical management and provide continuously usable services.

To utilize health lifelog data in real time, the application of artificial intelligence (AI) technology, capable of classifying large volumes of data and deriving meaningful results in real time, is essential. AI technology can serve as a critical decision-making tool for the early diagnosis of dementia. Various studies have demonstrated the extensive use of AI in real-time data collection and analysis [14]. These studies highlight the important role of AI technology in real-time data analysis and decision making. In a study utilizing lifelog data to predict diabetes and cardiovascular diseases, machine learning models demonstrated a precision of 97.1% and a recall of 96.2%, thereby validating the effectiveness of early diagnosis through lifelog data analysis [15]. Building on these findings, digital healthcare platforms that leverage lifelog data have continued to evolve, collecting and automatically analyzing individual health data to offer personalized health management. These platforms employ AI-based deep learning modules to perform real-time analyses, making them highly effective tools for managing chronic diseases [16]. Therefore, the application of AI technology to the real-time analysis of health lifelog data is considered a valid approach.

Dementia diagnosis is a complex process that requires specialized knowledge of various conditions and scenarios. However, based on the results of prior research, implementing an AI-based early dementia diagnosis prediction system by integrating health lifelog data with AI technology appears feasible. Therefore, in this study, we aim to develop an AI-based predictive algorithm that enables early dementia diagnosis using health lifelog data, serving as preliminary research for building an AI-based diagnostic system. The results of this study are expected to facilitate early dementia diagnosis, enabling appropriate and timely treatment, thereby slowing disease progression and improving patients’ quality of life.

## 2. Methodology

### 2.1. Participants

To achieve the objectives of this study, we utilized the “Wearable Lifelog Data for Dementia High-Risk Groups” provided by AI Hub, accessed on 26 June 2024. (https://www.aihub.or.kr/). AI Hub is an integrated AI platform operated by the National Information Society Agency of Korea, offering training data in six fields, including healthcare, to support AI service development. The wearable lifelog data for dementia high-risk groups were derived from raw data collected using healthcare wearable devices. These data underwent refinement and labeling processes for lifelog big data construction for each stage of dementia progression and included datasets indicating the probability of developing dementia, as assessed by an AI-based early prediction model. The dataset was collected from men and women aged 55 years residing in Gwangju Metropolitan City, based on precise diagnoses by specialists. A total of 300 participants were categorized into Cognitive Normal (CN), Mild Cognitive Impairment (MCI), and Dementia (Dem), and were equipped with ring-shaped wearable devices for data collection. Following data collection, participants with a wearable device usage period of less than 35 days were excluded through data preprocessing. The final distributed dataset included the cognitive function data of 174 participants, with 111 categorized as CN, 51 as MCI, and 12 as Dem.

### 2.2. Data Preprocessing and Variable Extraction

The Dem group’s relatively small sample size of 12 raises the model bias risk during training. Therefore, instead of classifying MCI and Dem separately, we combined the MCI and Dem data to classify them as a high-risk group for dementia. The training and valid data used in this study are summarized in Table 1.

In this study, lifelog data used to predict high-risk dementia groups were divided into sleep and gait data. The specific datasets, listed in Table 2, span approximately 35–120 days, and various metrics were collected daily. As the collection dates varied for each participant, we calculated the mean, standard deviation, maximum, and minimum values for each variable per participant to incorporate into the model training.

Specifically, unique participant lists were extracted from sleep and gait data using each participant’s email address as an identifier. For each participant, the mean, standard deviation, maximum, and minimum values of the variables were calculated and saved as separate Excel files. The training and valid data were saved separately to maintain distinct datasets for analysis.

Nonquantifiable variables (dates, etc.) in the gait lifelog data such as the five-minute activity log, activity start time, activity end time, and one-minute MET log were excluded from the analysis. Variables (dates, etc.) in the sleep lifelog data such as sleep start time, sleep end time, five-minute heart rate log, sleep state log, and five-minute heart rate variability log were excluded from the analysis.

Finally, StandardScaler was applied to standardize the training data. Data were preprocessed to ensure that the same scaler could be applied to the validation data. This approach was designed to enable early diagnosis of dementia using data standardized consistently with the training data. The scaler was saved along with the model, allowing new data not used in this study to be standardized according to identical criteria.

StandardScaler is one method used in data preprocessing for machine learning. It standardizes data by adjusting each feature to have a mean of 0 and a standard deviation of 1 [17]. This process can enhance the model’s performance and reduce the training time.

### 2.3. Data Augmentation

In this study, data augmentation was performed solely on the training data to improve the model’s performance. Collecting large amounts of data for machine learning research can be costly, particularly when using human data [18]. Data augmentation can be used effectively to enhance the model performance and prevent overfitting during data prediction [19]. We utilized models both with and without data augmentation. For this study, data augmentation was achieved by randomly adjusting the mean values of each measurement for participants within a ±10% range of the standard deviation, thereby increasing the training data size by 20 times. The Mean Value variable represents the measurement’s average value, and Standard Deviation refers to the measurement’s standard deviation. The Random Factor variable is a random number generated between −1 and 1 that scales the adjustment within the range of ±10% of the standard deviation. This method was applied repeatedly to increase the training dataset size by a factor of 20, ensuring substantial augmentation while maintaining the inherent statistical properties of the original data. Importantly, no augmentation was applied to the validation data in order to avoid potential issues that data augmentation might introduce during validation. The final training data used in this study after augmentation are presented in Table 3.
(1)Augmented Value=Mean Value+(Random Factor×0.1×Standard Deviation)

The final data used in this study after data augmentation are presented in Table 3.

### 2.4. Learning Model and Hyperparameters

In this study, prediction models were developed using Logistic Regression, Random Forest, LightGBM, and Support Vector Machine Classification. For datasets with defined features or limited data, traditional machine learning techniques are more effective than deep learning [20]. Additionally, to optimize the model’s hyperparameters, GridSearch, a technique to improve model performance in machine learning, was employed. This involves entering a list of hyperparameter values for a machine learning model, evaluating the performance for all possible combinations of these values, and identifying the best set of values [21]. The optimal hyperparameters for each model, determined using the GridSearch method, are listed in Table 4.

#### 2.4.1. Logistic Regression

Logistic Regression is a widely used machine learning technique that has been employed in various research areas. It is particularly effective in addressing binary classification problems by outputting probabilities between 0 and 1, which can then be used to predict the likelihood that a specific condition is true. For instance, in the context of dementia classification, a Logistic Regression model can predict whether a patient has dementia by providing a yes or no answer based on the calculated probability. Logistic Regression has traditionally been extensively utilized in the medical field, including applications such as dementia diagnosis, where it plays a crucial role in predicting patient outcomes. The primary reason for the widespread use of Logistic Regression lies in its ability to perform binary classification with a high degree of interpretability and efficiency. There is prior research demonstrating the effectiveness of Logistic Regression in predicting Alzheimer’s disease, achieving a high accuracy of 0.873 [22]. Additionally, this technique has been employed in various studies to predict disease risks, such as colorectal cancer [23] and diabetes [24]. Based on the results of these studies, it is evident that Logistic Regression is a commonly used method for disease prediction. Therefore, it was also adopted as the machine learning technique in this study. The specific formula for the Logistic Regression model is shown in Equation (2).
(2)$$-\frac{1}{m}\sum_{i=1}^{m}[y^{\left(i\right)}\log\left(h\left(z^{\left(i\right)}\right)\right)+(1-y^{\left(i\right)})log(1-h\left(z^{\left(i\right)}\right))$$

#### 2.4.2. Random Forest

Random Forest is an ensemble method that enhances predictive performance by combining multiple decision trees. Each tree in the model is trained independently, and the final prediction is determined through majority voting among the individual trees. Owing to this structure, Random Forest exhibits high stability and prediction accuracy, and it is particularly robust against overfitting. A decision tree operates by recursively partitioning the dataset into subsets based on various criteria. This partitioning process continues until no further predictions can be made, or until all data within a subset share the same target variable value. This recursive partitioning method is known as ‘Top-Down Induction of Decision Trees (TDIDT)’, and ultimately, the dependent variable Y is used as the target for classification. The vector v can be expressed by the following equation.
(3)$$\left(v,\;Y)=(x_{1},x_{2},\ldots ,x_{d},\;Y\right)$$

When training a Tree model, the process involves optimizing the parameters for each terminal and internal node, as well as the parameters of the node split function, to minimize an objective function defined based on the given data *v*, training set S0, and actual labels. Random Forest enhances model accuracy by utilizing an ensemble of these decision trees through the Bagging (bootstrap aggregation) method. Bagging involves repeatedly sampling the data (Bootstrap), training each model on these samples, and then aggregating the results (Aggregation) to produce the final prediction. In Random Forest, the combination of results from decision trees, each composed of different nodes, yields an optimized classification outcome.
(4)$$I_{G}\left(f\right)=\sum_{i=1}^{m}f_{i}\left(1-f_{i}\right)=\sum_{i=1}^{m}\left(f_{i}-f_{i}^{2}\right)=\sum_{i=1}^{m}f_{i}-\sum_{i=1}^{m}f_{i}^{2}=1-\sum_{i=1}^{m}f_{i}^{2}$$

The use of the Random Forest technique in the field of dementia diagnosis is justified by the complex and multidimensional nature of dementia-related data, which can be effectively handled by the multi-tree structure of this method. Studies have leveraged these advantages, applying the Random Forest technique to classify neuroimaging data related to Alzheimer’s disease [25]. Additionally, this method has been employed in various studies, such as those predicting cardiovascular diseases [26], demonstrating its utility in disease prediction. Given the results of these studies, it is evident that Random Forest is a widely used technique for disease prediction, which is why it was chosen as the machine learning method in this study. The specific formulas for the Random Forest technique are presented in Equations (3) and (4).

#### 2.4.3. LightGBM

LightGBM is a boosting framework that prioritizes high performance and speed, making it particularly useful for handling large-scale or complex high-dimensional data. Unlike traditional boosting algorithms that expand trees level by level, LightGBM grows trees leaf-wise. This approach allows the model to improve accuracy by preferentially splitting the leaf with the highest loss, thereby reducing overfitting. Additionally, LightGBM is well-suited for addressing data imbalance issues. This capability is especially valuable in research focused on disease prediction, such as dementia, where it effectively handles imbalanced datasets and facilitates early diagnosis and classification. For instance, studies have utilized LightGBM to develop models predicting the progression from mild cognitive impairment (MCI) to dementia, demonstrating high accuracy and efficiency [27]. Furthermore, LightGBM has been employed in various studies for disease prediction, including research on predicting thyroid disorders [28]. Given the outcomes of these prior studies, it is evident that LightGBM is a commonly used technique for disease prediction. Consequently, it was also selected as the machine learning method in this study.

#### 2.4.4. Support Vector Machine Classification

Support Vector Machine (SVM) is a model that establishes a decision boundary to classify data into two categories. When new data are input, the model analyzes the features of the data and classifies them into the category that corresponds to the side of the decision boundary with similar attributes. The performance of SVM improves as the margin between the decision boundary and the data increases, with this margin being referred to as the “Margin”. SVM enhances classification accuracy by securing a wider margin and strengthens the model’s reliability by removing outliers within the margin. The SVM algorithm can be described using a p-dimensional hyperplane, as expressed in Equation (5), which represents a line where $f\left(X\right)=0$:(5)$$f\left(X\right)=\beta_{0}+\beta_{1}X_{1}+\ldots +\beta_{p}X_{p}$$
(6)$$f\left(X\right)=0$$

The function *f*(*X*) classifies data points based on their position relative to the hyperplane: if *f*(*X_i*) is greater than 0, the data point is classified as Class 1; if *f*(*X_i*) is less than 0, it is classified as Class 2. Specifically, when *f*(*X_i*) > 0, the data point belongs to Class 1, and when *f*(*X_i*) < 0, it belongs to Class 2. Consequently, each data point is assigned a value of *Y_i*, which is either −1 or 1 depending on its class. The condition for determining that all data points are correctly classified is when the expression in Equation (7) is positive for all data points.
(7)$$Y_{i}\left(\beta_{0}+\beta_{1}X_{i1}+\ldots +\beta_{p}X_{ip}\right)>0$$

When drawing a hyperplane, data can be divided with various slopes; however, to create the most accurate classification model, it is essential to find the margin that maximizes the distance between the two classes. This process defines the optimal hyperplane. Therefore, it is crucial to find the value that maximizes this margin, as expressed in Equation (10), to optimize the model’s performance.
(8)Mβ0,β1,…,βp,ϵ1,…,ϵnMMaximize
(9)$$subject\;to\sum_{j=1}^{p}\beta_{j}^{2}=1$$
(10)$$Y_{i}\left(\beta_{0}+\beta_{1}X_{i1}+\ldots +\beta_{p}X_{ip}\right)\geq M(1-\epsilon_{1})$$
(11)$$\epsilon_{1}\geq 0,\sum_{i=1}^{n}\epsilon_{i}\leq C$$

The use of SVM in dementia classification is justified for several reasons. First, SVM is highly effective in handling high-dimensional data, making it well-suited for analyzing complex medical images such as MRI scans. Second, SVM has demonstrated high accuracy in predicting early stages of diseases like dementia [29], allowing for the construction of robust predictive models by integrating various clinical and imaging data. Additionally, SVM has been employed in a range of studies for disease prediction, including research on cardiovascular diseases [30]. Based on the outcomes of these previous studies, SVM is a widely used method for disease prediction, which is why it was also chosen as the machine learning technique in this study.

### 2.5. Model Evaluation Method

The performance of the prediction models was evaluated using six metrics: recall, precision, sensitivity, specificity, accuracy, and AUC. These performance indicators ranged from 0 to 1, with higher values indicating better model performance. The specific formula is as follows.
(12)$$Recall\;\left(Sensitivity\right)=\frac{True\;Positives\;(TP)}{True\;Positives\;\left(TP\right)+False\;Negatives\;(FN)}$$
(13)$$Precision=\frac{True\;Positives\;(TP)}{True\;Positives\;\left(TP\right)+False\;Positives\;(FP)}$$
(14)$$Specificity=\frac{True\;Negatives\;(TN)}{True\;Negatives\;(TN)+False\;Positives\;(FP)}$$
(15)$$Accuracy=\frac{True\;Positives\;\left(TP\right)+True\;Negatives(TN)}{Total\;number\;of\;cases}$$

### 2.6. Analysis Procedure

The research procedure was methodically organized into six stages to achieve the objectives of this study. Initially, wearable lifelog data from a high-risk dementia group was collected via AI Hub. This dataset included various physiological and behavioral indicators such as activity patterns, sleep cycles, and heart rate measurements. In the second stage, data preprocessing was conducted, which involved the removal of extraneous variables, computation of basic statistics, and standardization of each variable. This standardization ensured that the data were adjusted to a uniform scale, and basic statistics were employed to render the data in a format suitable for learning. The third stage involved segregating the preprocessed data into training and validation sets. Specifically, for the CN group, the data were divided in an 8:2 ratio for training and validation purposes, while for the MCI + Dem group, the data were split in a 9:1 ratio. This structured approach ensures a systematic processing of the data, setting a robust foundation for the subsequent analytical stages of the research. In the fourth stage, data augmentation was implemented to address issues of data scarcity and class imbalance. This step was crucial for enhancing the accuracy of the models and improving their performance across diverse conditions, underscoring the efforts to optimize data utilization. The fifth stage involved adjusting the hyperparameters of four predictive models. For this purpose, the GridSearch method was employed to systematically explore all combinations of hyperparameters to identify the configuration that yielded the best performance. Utilizing GridSearch significantly reduces the time researchers spend on manually testing combinations, thereby streamlining the model optimization process. In the final stage, the performance of the predictive models was evaluated using various metrics. Precision, recall, accuracy, and the area under the curve (AUC) were employed to thoroughly assess and analyze the predictive efficacy of each model. The specific research procedures are depicted in Figure 1.

## 3. Results

### 3.1. Results of Prediction Model Training Using Original Data (Gait)

This study aimed to predict high-risk dementia groups using lifelog data. As the first research outcome, prediction models were trained using the original gait lifelog data. The performance metrics for each trained model are detailed in Table 5. Although Logistic Regression, LightGBM, and Support Vector Machine performed well in terms of accuracy, Random Forest outperformed them all, with an AUC of 0.734 according to the ROC curve. However, sensitivity, which indicates the model’s ability to correctly identify actual dementia patients, was relatively low at 0.429. Detailed confusion matrices for each model are provided in the Appendix A.

### 3.2. Results of Prediction Model Training Using Original Data (Sleep)

As the second research outcome, prediction models were trained using the original sleep lifelog data. The performance metrics for each trained model are listed in Table 6. Both the accuracy and AUC scores indicated that the Support Vector Machine model performed the best. However, sensitivity was relatively low at 0.429. Detailed confusion matrices for each model are provided in the Appendix A.

### 3.3. Results of Prediction Model Training Using Augmented Data (Gait)

As the third research outcome, the prediction models were trained using augmented gait lifelog data. The performance metrics for each trained model are listed in Table 7. Although the Support Vector Machine model had the highest accuracy, the Random Forest model demonstrated superior performance when considering the AUC. Notably, performance improved compared to the pre-augmentation AUC of 0.734, with sensitivity increasing from 0.429 to 0.571. Detailed confusion matrices for each model are provided in the Appendix A.

### 3.4. Results of Prediction Model Training Using Augmented Data (Sleep)

As the fourth research outcome, prediction models were trained using the augmented sleep lifelog data. The performance metrics for each trained model are listed in Table 8. Although the Support Vector Machine model had the highest accuracy, the Logistic Regression model demonstrated better performance when considering the AUC. Sensitivity also improved from 0.429 to 0.571. Detailed confusion matrices for each model are provided in the Appendix A.

## 4. Discussion

This study aimed to develop an algorithm for the early diagnosis of high-risk dementia groups among pre-older adult individuals using AI. By leveraging health lifelog data that are easily accessible in everyday life, this study suggests a significant potential to reduce reliance on expensive medical equipment and specialized medical professionals required by traditional diagnostic methods. This approach could be especially applicable in areas with limited access to healthcare or in economically disadvantaged environments, offering the potential to improve public health quality. The key findings of the study are summarized as follows.

First, participants’ cognitive function in this study was categorized into three groups: CN (111), MCI (51), and Dem (12). This led to an imbalance in the data between groups. Although we attempted to address this issue, it did not resolve the imbalance between MCI and Dem or the gender distribution. Consequently, patients with MCI and Dem were combined and labeled as the high-risk dementia group. Data imbalance is a common limitation in dementia prediction studies. For instance, in previous studies, the Synthetic Minority Over-sampling Technique (SMOTE) was proposed to artificially augment data for minority groups in machine learning-based research [31]. This technique helps balance datasets by generating examples of the minority class.

In healthcare-related research, data augmentation has been employed to address the issue of insufficient sample sizes, such as in studies focused on classifying human body types using deep learning techniques [32]. Additionally, efforts to build AI systems aimed at preventing doping among athletes have also utilized data augmentation to compensate for limited sample sizes [33]. Following these examples from various previous studies, this research applied data augmentation to tackle the problem of data imbalance. However, in the long term, future research will need to employ more precise classification techniques and expand data collection across diverse populations.

Second, this study aimed to develop an algorithm for the early diagnosis of high-risk dementia groups using original lifelog data. Although the algorithm achieved a maximum accuracy of 0.879, the sensitivity for correctly classifying actual dementia cases was low at 0.429. This raises questions about whether this algorithm is optimal for classifying dementia, which could be a topic for discussion among researchers. However, the accurate and quick prediction of patients with dementia is crucial for its management and treatment. Precise and rapid diagnosis plays a decisive role in establishing appropriate treatment and management plans, which are essential for maintaining a patient’s quality of life and slowing the progression of dementia.

Using lifelog data collected from daily life for proactive early diagnoses of dementia would considerably aid the initiation of appropriate treatment at an early stage. For example, in previous studies, a machine learning-based system using lifelog data has detected abnormal behaviors in dementia patients. This system demonstrated the potential to monitor patient conditions in real time and identify issues early [34]. This study underscores the importance of early diagnosis and continuous monitoring in dementia management and suggests how technological approaches can improve patient care. The use of lifelog data has expanded in various fields, not only in dementia research. These data are increasingly being applied in the context of early and proactive diagnosis, where speed is often prioritized over precision. In this regard, low-cost wearable devices play a crucial role, serving as important tools for rapid data collection and analysis. The data collection device used in this study aligns with this approach. For instance, in previous research, the authors developed a low-cost, autonomous wearable device designed to track Alzheimer’s patients. The device uses GPS and geofencing technology to monitor the patient’s location in real time and sends alerts when the patient exits a designated safe zone [35]. Such low-cost devices help alleviate the burden on patients and their families and can be effectively utilized in regions with limited access to healthcare.

Third, we addressed the data imbalance issue by performing data augmentation. Numerous studies have proposed data augmentation as a solution to data imbalance problems [32]. The results of this study support previous findings, showing improved performance in sensitivity after data augmentation, indicating that the ability to classify actual dementia cases as dementia improved. An increase in sensitivity implies a better identification of actual dementia cases, potentially leading to a more accurate diagnosis. Therefore, we hope that future studies will continue to explore various techniques to enhance sensitivity. Furthermore, future studies should investigate how these techniques can be applied in clinical settings to contribute to high-quality research capable of early dementia prediction.

## 5. Conclusions

This study aimed to develop a predictive algorithm using AI technology for the early diagnosis of high-risk dementia groups among pre-older adult individuals. Early diagnostic methods that utilize health lifelog data aim to overcome the limitations of traditional diagnostic methods. The results suggest that effectively utilizing lifelog data, which can be easily collected from daily life, not only enhances the accessibility of dementia diagnosis and enables the efficient use of medical resources but also plays a crucial role in delaying the progression of dementia.

This study focused on improving model accuracy and sensitivity by applying data augmentation techniques to overcome the limitations of previous studies, such as data imbalance issues. The improvement in sensitivity after data augmentation can enhance the reliability of AI-based diagnostic systems. Nevertheless, future research should address the issue of data imbalance, as well as efforts to improve sensitivity.

Finally, the approach used in this study suggests the potential for application not only in dementia diagnosis but also in the early diagnosis of various health conditions. The integration of AI and healthcare technology could lead to more precise and personalized medical services, further improving the quality of public health. Future research should aim to enhance the model’s predictive power using more diverse data and advanced analytical techniques and explore its applicability in real clinical settings.

## Figures and Tables

**Figure 1 healthcare-12-01872-f001:**
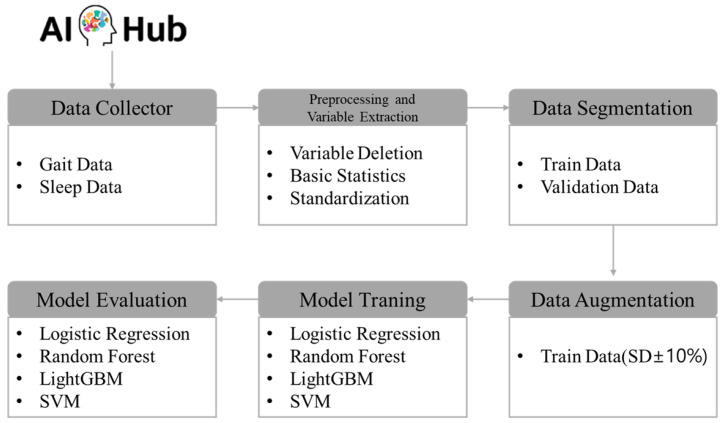
Analysis procedure.

**Table 1 healthcare-12-01872-t001:** Dataset used in this study.

Classification	CN	High-Risk Group for Dementia (MCI + Dem)
Train	85	56
Valid	26	7

**Table 2 healthcare-12-01872-t002:** Description of gait and sleep lifelog data.

Classification	Gait Data		Sleep Data	
	Variable	Description	Variable	Description
1	activity_average_met	Average Daily Physical Activity Intensity	sleep_awake	Wake Time
2	activity_cal_active	Daily Activity Calories	sleep_breath_average	Average Respiratory Rate per Minute
3	activity_cal_total	Total Daily Calorie Expenditure	sleep_deep	Deep Sleep Time
4	activity_daily_movement	Daily Distance Moved	sleep_duration	Total Sleep Time
5	activity_high	High-Intensity Activity Duration	sleep_efficiency	Sleep Efficiency
6	activity_inactive	Inactivity Duration	sleep_hr_average	Average Heart Rate per Minute
7	activity_inactivity_alerts	Inactivity Alarm Frequency	sleep_hr_lowest	Low Heart Rate per Minute
8	activity_low	Low-Intensity Activity Duration	sleep_is_longest	Confirmed Sleep Presence
9	activity_medium	Moderate-Intensity Activity Duration	sleep_light	Light Sleep Time
10	activity_met_min_high	Daily High-Intensity Physical Activity Intensity	sleep_midpoint_at_delta	Sleep Midpoint Time (Delta)
11	activity_met_min_inactive	Daily Inactivity Physical Activity Intensity	sleep_midpoint_time	Sleep Midpoint Time
12	activity_met_min_low	Daily Low-Intensity Physical Activity Intensity	sleep_onset_latency	Sleep Latency
13	activity_met_min_medium	Daily Moderate-Intensity Physical Activity Intensity	sleep_period_id	Sleep Identification ID
14	activity_non_wear	Non-wear Duration	sleep_rem	REM Sleep Duration
15	activity_rest	Rest Duration	sleep_restless	Toss and Turn Rate
16	activity_score	Activity Score	sleep_rmssd	Average Heart Rate Variability
17	activity_score_meet_daily_targets	Activity Goal Achievement Score	sleep_score	Overall Sleep Score
18	activity_score_move_every_hour	Hourly Activity Maintenance Score	sleep_score_alignment	Sleep Timing Score
19	activity_score_recovery_time	Recovery Time Score	sleep_score_deep	Deep Sleep Score
20	activity_score_stay_active	Activity Maintenance Score	sleep_score_disturbances	Sleep Disturbance Score
21	activity_score_training_frequency	Exercise Frequency Score	sleep_score_efficiency	Sleep Efficiency Score
22	activity_score_training_volume	Exercise Score Training Volume	sleep_score_latency	Sleep Latency Score
23	activity_steps	Daily Step Count	sleep_score_rem	REM Sleep Score
24	activity_total	Total Activity Duration (minutes)	sleep_score_total	Sleep Duration Contribution Score
25	-	-	sleep_temperature_delta	Skin Temperature Deviation (Delta)
26	-	-	sleep_temperature_deviation	Skin Temperature Deviation
27	-	-	sleep_total	Sleep Time

**Table 3 healthcare-12-01872-t003:** Dataset utilized in this study after data augmentation.

**Classification**	**CN**	**High-Risk Group for Dementia (MCI + Dem)**
Train	1700	1120
Valid	26	7

**Table 4 healthcare-12-01872-t004:** Optimal hyperparameters for each machine learning model.

Classification	Model	Hyperparameters	
		Gait Data	Sleep Data
1	Logistic Regression	c = 0.01	c = 0.01
2	Random Forest	Max_depth = 15	Max_depth = 15
		n_estimators = 200	n_estimators = 50
		random_state = 2024	random_state = 2024
3	LightGBM	learning_rate = 0.01	learning_rate = 0.01
		n_estimators = 50	num_leaves = 15
		num_leaves = 15	
4	Support Vector Machine Classification	c = 1	c = 1
		gamma = 0.01	gamma = 1
		probability = True	probability = True

**Table 5 healthcare-12-01872-t005:** Results of prediction model training using original data (gait).

Model	Recall/Sensitivity	Precision	Specificity	Accuracy	AUC
Logistic Regression	0.429	1.000	1.000	0.879	0.621
Random Forest	0.429	0.429	0.846	0.758	0.734
LightGBM	0.429	1.000	1.000	0.879	0.643
Support Vector Machine Classification	0.429	1.000	1.000	0.879	0.681

**Table 6 healthcare-12-01872-t006:** Results of prediction model training using original data (sleep).

Model	Recall/Sensitivity	Precision	Specificity	Accuracy	AUC
Logistic Regression	0.429	1.000	1.000	0.879	0.780
Random Forest	0.429	0.429	0.846	0.758	0.745
LightGBM	0.429	0.750	0.962	0.848	0.769
Support Vector Machine Classification	0.429	1.000	1.000	0.879	0.786

**Table 7 healthcare-12-01872-t007:** Results of prediction model training using augmented data (gait).

Model	Recall/Sensitivity	Precision	Specificity	Accuracy	AUC
Logistic Regression	0.429	0.375	0.808	0.727	0.604
Random Forest	0.571	0.500	0.846	0.788	0.808
LightGBM	0.429	0.600	0.923	0.818	0.736
Support Vector Machine Classification	0.429	1.000	1.000	0.879	0.764

**Table 8 healthcare-12-01872-t008:** Results of prediction model training using augmented data (sleep).

Model	Recall/Sensitivity	Precision	Specificity	Accuracy	AUC
Logistic Regression	0.571	0.571	0.885	0.818	0.802
Random Forest	0.571	0.500	0.846	0.788	0.734
LightGBM	0.000	0.000	1.000	0.788	0.544
Support Vector Machine Classification	0.429	1.000	1.000	0.879	0.786

## Data Availability

This research (paper) used datasets from ‘The Open AI Dataset Project (AI-Hub, S. Korea)’. All data information can be accessed through ‘AI Hub, accessed on 26 June 2024 (www.aihub.or.kr)’.

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
