# Peer review of "Development of an AI-Based Predictive Algorithm for Early Diagnosis of High-Risk Dementia Groups among the Elderly: Utilizing Health Lifelog Data"

_healthcare, 2024, doi:10.3390/healthcare12181872_

Round 1
Reviewer 1 Report
Comments and Suggestions for Authors
Appreciate the authors motive to draft a manuscript entitled “Development of an AI-Based Predictive Algorithm for Early Diagnosis of High-Risk Dementia Groups among the Elderly: Utilizing Health Lifelog Data”. The aim of the work is to identify a predictive algorithm for the early diagnosis of dementia in the high-risk group of older adults using artificial intelligence. The data source AI HUB is used and traditional methods of machine learning is implemented in the study.
The following are some of my observations/ Suggestions for further revision of manuscript.
1. In abstract section, the obtained accuracy can be mentioned in classifying dementia cases.
2. In Introduction section, the authors have not discussed on the different type of the machine learning methods used in the earlier research or related works. The readers will not have a clear idea on the used methods and therefore the gap analysis is not transparent to them. (This section needs clear explanation of the concepts used and the output obtained in each model delivered by early researchers)
3. In the Methodology section, why limited participants were taken into account? Preprocessing methods and Data augmentation methods (i.e) filters, transformation etc along with their mathematical relations are not elaborated.
4. In the learning models, the logistic Regression, Random Forest, LightGBM, and Support Vector Machine Classification algorithms procedures or flow is missing.
5. Fig.1 is general flow, the architecture and flow diagram should give overall clarity about the research work conducted including submodules.
6. What percentage of dataset is used for training, testing and validation?
7. There is no mathematical expression for calculating Recall, Precision, Sensitivity, Specificity, Accuracy, AUC. Why ROC is not preferred?
8. Why authors have compared with original data and data augmentation? In my opinion, it is not needed; Suggested that the data augmentation is part of the work to enhance the dataset.
9. Why not authors have a comparative study on using different ml methods?
10. The results section is missing with the number of Epoch, testing and validation graph.
11. AI Proactive algorithms innovation is missing. In the manuscript, existing methods are used for comparison.
12. In what way references 20,21,22 are related to the proposed work.
General observation:
1. Across the script there arises a confusion in using the words, AI Technologies, ML Methods, DL Methods. Authors should use the appropriate terms when it is needed only. There is no clarity in the script and understanding of AI Technologies, ML Methods, DL Methods from the reader point of view.
2. Computational environment is missing.
3. References shall be further strengthen with more related works.
Comments on the Quality of English LanguageNeeds to be improved.
Reviewer 2 Report
Comments and Suggestions for Authors
The introduction and methods sections are well-described. However, I have a few suggestions for improvement:
Figure 1: In the model evaluation section, it would be more appropriate to include classification metrics such as accuracy or recall rather than listing the algorithms used to create the models. This would provide a clearer understanding of the model performance. Additionally, if I understood correctly from your earlier description, the models were created with and without data augmentation. The current diagram suggests that the models were only based on augmented data, so this should be clarified.
Results: This section needs improvement. The results currently focus on the training set, but it is essential to include external validation with a test set to ensure the robustness of the findings.
I hope these suggestions are helpful in strengthening the paper.
Comments on the Quality of English LanguageMinor editing of language is required
Reviewer 3 Report
Comments and Suggestions for Authors
This study focuses on predicting dementia using data gathered from patient activity information collected via a wearable device. Several algorithms were applied, with their parameters optimized through grid search. Additionally, the potential effects of data augmentation were examined.
While the study addresses an interesting topic relevant to academia, it lacks novelty. From a technical perspective, well-known algorithms were used, and only basic model optimization techniques were applied. Moreover, the dataset splitting and validation methods appear to have been incorrectly implemented. From an application standpoint, the results are insufficient. It seems that none of the models performed better than a baseline model that simply predicts the most frequent category.
- Introduction
The introduction is clear, rigorously defining the problem and introducing the potential solutions explored in this manuscript. However, it would have been beneficial to include more detailed information on the types of data typically collected in LifeLog studies.
- Methodology
- Dataset Splitting: The manuscript lacks clarity on how the dataset was split into training and validation sets.
- Table 2: This table is captioned "Description of Gait and Sleep LifeLog Data," but only the variable names are provided. It would enhance the understanding of the dataset to include a brief description of each variable. Given the potential length of the table, this information could be included in an appendix.
- Excluded Variables: Lines 140 to 145 mention a set of excluded variables, but the rationale behind their exclusion is not clear.
- Total Number of Variables: The total number of variables used for training should be specified.
- Data Scaling: The paragraphs between lines 147-155 should be combined. First, define what standard scaling is, and then clarify how it was applied. It is important to specify which data were used to compute the mean and standard deviation.
- Data Augmentation: The data augmentation method used should be clarified. Including some equations to explain how new data were generated would help in understanding the method.
- Model Search Methodology: Section 2.4 explains the model search methodology but does not specify which data were used to evaluate each model. Was the validation set used for the GridSearch evaluations?
- Inconsistency in Terminology: Figure 1 states that the data is split into Train and Test datasets; however, the manuscript frequently uses the term "validation set." This inconsistency should be addressed.
- Test Set Augmentation: The test set appears to have been augmented as well. Why was this done? The test set should represent real-world data, and it is important to question whether the real-world incidence of dementia is balanced.
- Results
- Metric Consistency: The tables show columns for both recall and sensitivity, but the values are identical in every row. Recall and sensitivity are the same metric, so presenting them separately is redundant.
- Analysis of Results: A more in-depth analysis of the results is necessary. The recall of 0.429, repeated in most of the models, suggests that the models are not learning effectively. Confusion matrices could better highlight this issue.
- References
- Missing References: The manuscript lacks references to other studies that use machine learning to predict dementia. There is existing literature on this topic that should be cited to provide context and contrast to this work.
Reviewer 4 Report
Comments and Suggestions for Authors
Reviewer’s comments.
Dear Authors,
As a reviewer, these are some modifications and recommendations for enhancing the article "Development of an AI-Based Predictive Algorithm for Early Diagnosis of High-Risk Dementia Groups among the Elderly: Utilizing Health Lifelog Data."
1. The abstract is well-organised, however the primary findings should be more clearly stated. Consider summarising the important findings and consequences more succinctly. Avoid employing excessive technical jargon. Simplify words like "LightGBM" and "Support Vector Machine" for a wider audience.
2. Although the introduction gives a good overview, it would be helpful to go into greater information about current approaches to early diagnosis of dementia and how the suggested approach differs from them. The authors should ensure that every assertion, particularly those using statistical data is backed up by current references. There are instances in the introduction where citations are missing. Ensure to cite the 5th paragraph in the introduction.
3. One drawback is the limited sample size (n=12) for the Dementia (Dem) group. This needs to be acknowledged, and a more thorough discussion of the factors that led to this tiny sample should ensue. One way to improve the description of data augmentation would be to provide the rationale behind the selection of particular procedures. Mention any possible drawbacks or dangers of data augmentation as well, such as the introduction of fictitious patterns that might not exist in data from the real world. The ethical issues of using lifelog data, particularly in vulnerable populations, should be explored even when an IRB review is waived.
4. Although Tables 5-8 contain a lot of information, the narrative might better guide the reader through these findings. If you want to know why some models performed better or worse, consider including an additional narrative. One important problem is that certain models have low sensitivity (0.429). The possible impact on practical application should be highlighted and this should be covered in greater detail. It is necessary to compare the outcomes with current models or procedures. In comparison to the industry's current norms, how well does the suggested model perform?
5. The topic of data imbalance is briefly discussed, but it should go into further detail on other restrictions, like the possibility of overfitting, the results' generalisability, and the dependence on certain datasets. More in-depth discussion should be held regarding the possible wider ramifications of utilising AI to diagnose dementia, including ethical concerns. The necessity for transparency in AI-based decision-making and possible biases in AI systems are examples of this.
6. More specific recommendations for future study areas, including how to overcome the noted limitations, should be made in the conclusion. Increasing the dataset size, enhancing sensitivity, and using the model on various populations are a few examples. More information should be provided about how this research might be used in clinical settings. How can this approach be incorporated into the current systems used by healthcare providers?
7. Given how quickly AI and healthcare research are developing, make sure all references are current. Perhaps more recent research should be used in place of some of the older references. Inspect the manuscript for uniformity in referencing style. It is important that all references use the same style.
8. The manuscript is well -written in general, but some instances revealed places where language could be streamlined for audience understanding. The authors should revisit the complex sentences to enhance readability. Furthermore, the authors should verify the formatting style particularly for figures and tables that comply with the journal’s guidelines.
Comments on the Quality of English LanguageThe article's English is typically of high quality, with mostly proper grammar and suitable terminology. To improve readability and clarity, there are, nevertheless, certain areas that need work. For easier comprehension, some sentences that are verbose or complex should be shortened. Furthermore, employing more active voice, minimising repetitive word choices, and making sure that paragraph transitions are seamless could all enhance the text's overall engagement and flow. The article's professionalism could be further increased by making small formatting and punctuation changes. Overall, even if the language is appropriate for an academic readership, these edits would improve its clarity and concision.
Round 2
Reviewer 1 Report
Comments and Suggestions for Authors
Appreciate the efforts taken to improve the manuscript
Comments on the Quality of English LanguageProof-Read Required
Author Response
Dear Reviewer 1,
I hope this message finds you well.
In response to your feedback, we have further proofread the manuscript to address the minor English language issues you pointed out. Additionally, we have reviewed and made improvements to the clarity and presentation of the introduction, research design, methods, results, and conclusions.
We are confident that these revisions have enhanced the manuscript and addressed the areas you highlighted. Should you have any further comments or suggestions, please feel free to let us know.
Once again, thank you for your valuable feedback and for appreciating our efforts to improve the manuscript. We look forward to your response.
Reviewer 2 Report
Comments and Suggestions for Authors
Dear authors,
thank you for the changes you have made
Author Response
Dear Reviewer 2,
I hope this message finds you well.
In response to your comments, we have taken additional steps to further improve the manuscript and ensure it meets the highest standards. We believe the revisions have strengthened the overall quality and clarity of our work.
Should you have any further suggestions or concerns, please feel free to share them, and we will be happy to address them.
Once again, thank you for your valuable feedback and for recognizing the improvements in the manuscript.
Reviewer 3 Report
Comments and Suggestions for Authors
Data Splitting Methodology:
You mentioned using an 8:2 ratio to split the original dataset, but the train and validation sets (85:26 and 56:7) do not correspond to this ratio. The data splitting methodology remains unclear. How was the original dataset divided? Was the split based on patients, time series, or another method? Was the division done randomly, or was another criterion applied? This needs to be clearly explained.
Figure 1 Inconsistency with Test Data:
The inconsistency with the test data in Figure 1 persists. Where is the test set derived from? Does a test set exist? If so, how many samples are included in the test set, and what is its distribution? These questions remain unanswered.
Did you use the validation data both to optimize the hyperparameters and to obtain the final results?
Test Set Augmentation:
The authors mentioned that their approach is supported by numerous precedents in the literature, particularly regarding the use of data augmentation for imbalanced datasets. While I agree that data augmentation is a valid strategy in this context, the key question has not been addressed: why was data augmentation applied to the test set? The test set should ideally represent the distribution of real-world data; otherwise, the results will not reflect real-world performance and could be misleading.
Metric Consistency:
The authors stated that they would use consistent terminology, but the table still presents both metrics. If it is important to retain both terms, I suggest specifying this clearly in the column names, for example, "Recall/Sensitivity."
Data Augmentation and Imbalance:
In the discussion section, the authors stated, "Third, we addressed the data imbalance issue by performing data augmentation." However, after data augmentation, the training and validation sets remain imbalanced. The common practice is to apply data augmentation to balance the dataset by generating more samples from the underrepresented categories. It seems inconsistent to apply augmentation without addressing the imbalance. Why didn’t the authors follow this approach? This needs to be clarified.
Results:
I believe it is not specified which class (CN or MCI+Dem) is considered the positive class. This distinction is crucial for accurately computing and interpreting the metrics.
In my opinion, there appears to be an underlying issue with either the data or the methodology. As I mentioned in the previous review, the results suggest that the models have not learned effectively. The authors state that accuracy is high, but accuracy is not an appropriate metric for imbalanced datasets, as it can be misleading. This is a key concern that should be addressed.
I requested the confusion matrices to assess the exact number of samples correctly classified in each category. However, these matrices have not been provided.
